# Evaluation of Biological Response of Lettuce (*Lactuca sativa* L.) and Weeds to Safranal Allelochemical of Saffron (*Crocus sativus*) by Using Static Exposure Method

**DOI:** 10.3390/molecules24091788

**Published:** 2019-05-08

**Authors:** Hossein Mardani, John Maninang, Kwame Sarpong Appiah, Yosei Oikawa, Majid Azizi, Yoshiharu Fujii

**Affiliations:** 1Department of International Environmental and Agricultural Sciences Tokyo University of Agriculture and Technology, Fuchu Campus, 2N405, 3-5-8, Saiwai-cho, Fuchu, Tokyo 183-8509, Japan; ksappiah90@gmail.com (K.S.A.); yosei@cc.tuat.ac.jp (Y.O.); 2Center for Global Communication Strategies (CGCS) College of Arts and Sciences, The University of Tokyo, 3-8-1 Komaba, Meguro-ku, Tokyo 153-8902, Japan; johnmaninang3@yahoo.com; 3Department of Horticulture, Faculty of Agriculture, Ferdowsi University of Mashhad, Mashhad 9177948974, Iran; azizi@um.ac.ir

**Keywords:** allelopathy, allelochemicals, catalase activity, volatile organic compounds, safranal

## Abstract

Safranal, the main volatile chemical of Saffron (*Crocus sativus*) was studied to estimate its allelopathic effects on the photosynthetic pigment chlorophyll, leaf electrolyte leakage, fresh weight, catalase (CAT), and peroxidase (POX) activity of the test plant Lettuce (*Lactuca sativa*). In this study, the effective concentration (EC_50_) of safranal on CAT was estimated to be 6.12 µg/cm^3^. CAT activity was inhibited in a dose-dependent manner by the increase in the safranal concentration while POX activity was increased. Moreover, Safranal caused significant physiological changes in chlorophyll content, leaf electrolyte leakage, and fresh weight of several weed species with *Lolium multiflorum* being the most sensitive. Furthermore, 5 µM Safranal showed significant inhibitory activity against dicotyledonous in comparison to the monocotyledons under greenhouse conditions. The inhibition of the CAT by safranal was similar to those of uncompetitive inhibitors, and therefore the decline in carbon fixation by plants might be the mechanism behind the inhibitory activity of safranal.

## 1. Introduction

Volatile organic compounds (VOCs) are the dominant group of bioactive compounds that are often involved in plant defence mechanisms and as chemical attractants of plant pollinators [1]. Allelopathy is a biological phenomenon caused by the release of plant-produced toxins or stimuli (allelochemicals) from aerial parts into the phyllosphere or from underground parts into the rhizosphere to influence the growth, survival, or reproduction of other organisms [2]. The search for environmental friendly and economical weed control options has increased the quest for exploring the allelopathic phenomenon by using allelochemicals as bio herbicides in agricultural systems [3,4,5]. Similarly, interest in the use of natural phytochemicals as a resource for developing pharmaceutical and agricultural products has increased dramatically in recent years. This situation has been triggered due to the evolution of herbicide resistance in many weeds and their negative effect on environment. [6]. Many plant species, including medicinal plants, have the potential to produce and release biologically active compounds into the environment that may have phytotoxic effect(s) on other plants [7,8,9]. This is most likely due to the high content of essential oils in the medicinal plants species [10]. Essential oils, which primarily are a mixture of VOCs have been studied for their weed control potential [11]. However, individual constituents of VOCs have been less studied for their potentials in alternative weed management strategies. This is mainly due to the high volatility and low solubility of the VOCs [12]. Some volatile bioactive compounds namely safranal and octanal identified in *C. sativus* and *Heracleum sosnowskyi*, respectively, were reported with strong inhibitory activity on plant growth and development [13,14].

*C. sativus*, or saffron, is a renowned spice in traditional medicine in Iran. It is known to be the most expensive spice in the world, and its documented cultivation in Iran goes back to 1000 BC [15]. Safranal (2,6,6-trimethyl-1,3-cyclohexadiene-1-carboxaldehyde), the main constituent of Saffron volatile compounds is known for its protective effects on different markers of oxidative damage of mammalian cells [16] and neuropsychological effects [17]. As for its pharmacological properties, it has been reported that safranal IC_50_ against cell proliferation and induced cell apoptosis of mouse neuroblastoma N2A cell line was 11.1 and 23.3 µg/mL [18]. The dose-dependent cancer-cell-killing activity of safranal (0.01e3 mM) against Hela and MCF7 cells has been previously estimated [19]. The oral application of Saffron extract could increase antioxidant enzymes activity and therefore down-regulate the Reactive Oxygen Species (ROS) generation in the mice skin, which might reduce skin papillomas [20]. Although the two main components of the Saffron pigments (crosin and picrocrosin) are well known for their antioxidant properties, the true effect of safranal on antioxidant activity is not well understood [16,20,21,22].

In plant science, when developing a new herbicide (or pesticide), it is important to understand the dose-response where the target plant shows the desired physiological changes. Numerous studies on herbicide toxicity on different organisms including fish, insects, and plants, have pointed out the negative effect of herbicides [23,24]. It has been reported that the application of some herbicides such as simetyrine generates ROS—including hydrogen peroxide (H_2_O_2_) and superoxide (O^−^_2_) in plants [25]. Therefore, any change or interference in the activity of ROS scavenging enzymes could affect the growth or survival of plants. Alterations in chlorophyll content (critical for carbon fixation), electrical conductivity (as a sign of cell wall disruption), and fresh weight can significantly affect plants survival. Many of the known allelochemicals act as plant growth inhibitors by initiating oxidative stress in plants [26]. It has been known that safranal and its analogues can alter or inhibit the activities of various enzymes, including F1Fo ATP and αR283D ATP synthase in bacteria [27]. In addition, safranal may act as specific inhibitor for many enzymes, which makes it appealing for developing safer drugs and agrochemicals [28]. Despite the recent attention towards the pharmaceutical potential of safranal and its activity, to date, dose-response and the molecular mechanisms by which plants perceive safranal remains ambiguous.

Therefore, this study was aimed to gain insight into the mechanism of action of safranal and evaluation of its potential for weed management by testing its effects on plant ROS scavenging enzymes (catalase (CAT) and peroxidase (POX)) of *L. sativa*. We also aimed to examine the dose and biological responses of some common weeds species to the application of safranal.

## 2. Results and Discussion

### 2.1. Physiological Response of L. sativa to the Volatile Safranal

The inhibitory activity of safranal on physiological characteristics of plants was first evaluated using *L. sativa* as a test plant. The chlorophyll content and the fresh weight of *L. sativa* seedlings were measured after being exposed to safranal using a chlorophyll meter (SPAD-502Plus, Konica Minolta Co., Tokyo, Japan). Figure 2A,B show that both the chlorophyll content and the fresh weight decreased significantly after 48 h of exposure to safranal in a dose-dependent manner. However, in comparison with fresh weight, the chlorophyll content of *L. sativa* leaf was more sensitive to the different safranal concentrations. The chlorophyll content also significantly decreased by approximately four folds in comparison to the control. Different environmental stresses such as salinity, drought, organic contaminants, herbicides, UV light, and heavy metals can cause physiological changes and increase oxidative stress in plants [29,30]. Allelochemicals also stimulate some stress responses including the production of H_2_O_2_ and other ROS that could result in growth reduction [26]. Decreasing chlorophyll content is a common phenomenon observed in plants exposed to environmental stresses such as herbicides or allelochemicals [29]. Accumulation of ROS, and therefore, an increase in chlorophyll scavenging activities due to the effect of safranal can be a reason for chlorophyll decline in this experiment. Furthermore, due to the decrease in chlorophyll content, the efficiency of carbon fixation decreased, which resulted in a lower fresh weight of seedlings (Figure 2A). The weight loss in plants due to the reduction in carbon fixation is usually seen when plants are faced with biotic and abiotic stress, and in extreme cases could result in cell death [31].

### 2.2. The Effect of Safranal on the Activity of CAT and GPOX, and Accumulation of Ros in Plant Tissues 

#### 2.2.1. Accumulation of ROS in Plant Tissues

The 3, 3′-Diaminobenzidine (DAB) staining method is a strong tool for the localization of ROS in plants and has been used in numerous physiological studies [32]. DAB creates a brown colour when reacting with the accumulated H_2_O_2_ in plant tissue [33,34]. The brown colour derived from the polymerization reaction of DAB and H_2_O_2_ in this experiment was observed in the entire plantlets body (Figure 1A). In the present study, H_2_O_2_ in *L. sativa* leaf tissue increased significantly because of the safranal induced stress. After 48 h of exposure, control plantlets generated a lower level of H_2_O_2_ compared to plantlets of safranal treatments. The maximum brown colour was observed in safranal treatments with higher concentrations. However, the observation for visualization with Nitroblue tetrazolium (NBT) method for localization of superoxide (O^−^) in the leaf tissue showed a lower generation of O^−^. Figure 2B shows a slight blue colour of NBT staining in the treated plants, indicating a lower level of superoxide in the seedlings. Interestingly, 20 µM safranal caused intense bubble formation and tissue disruption in the leaf tissue, indicating that the accumulation of ROS (H_2_O_2_) due to safranal could result in severe membrane and cell wall disruption (Figure 1C). ROS are known to cause intense damage to the cell membrane [33]. Accumulation of ROS due to the application of active compounds, whether they are artificial or natural, has been reported as one of the main mechanisms of action of herbicides [25].

#### 2.2.2. CAT and GPOX Enzyme Assay Results

The safranal treatment resulted in significant enzymatic activities as compared to the control. The application of 5 µM safranal decreased the activity of CAT by more than three-folds (Figure 2D). The stimulation of CAT activity due to the production of O^−^_2_ and H_2_O_2_ have been reported as the mode of action of some bio herbicides [35,36]. It is possible that safranal may cause dramatic antioxidant inhibitory activity. However, in comparison with the 20 µM concentration, the GPOX activity remained high by increasing the safranal concentrations (5 and 10 µM) (Figure 2C). These results indicate that safranal may act as an uncompetitive enzyme inhibitor for CAT. The antioxidant activity of safranal in low dosage on mammalian cells has been well documented [37]. However, the effect of safranal and other Saffron chemical compounds on the growth and survival of plant tissues have not been reported. Ideally, when plants face a biotic or abiotic stress, ROS accumulation such as H_2_O_2_ and O^−^_2_ in leaf tissue rapidly increase and, therefore, plant antioxidant enzymes are triggered [24,29,34]. However, in the present study, quantification of CAT activity and the generation of H_2_O_2_ showed that despite the rapid generation of H_2_O_2_, safranal greatly inhibited the activity of CAT. In contrast, GPOX activity increased, and subsequently a lower concentration of O^−^_2_ was accumulated in the leaf tissue (Figure 2C,D).

**Figure 2 molecules-24-01788-f002:**
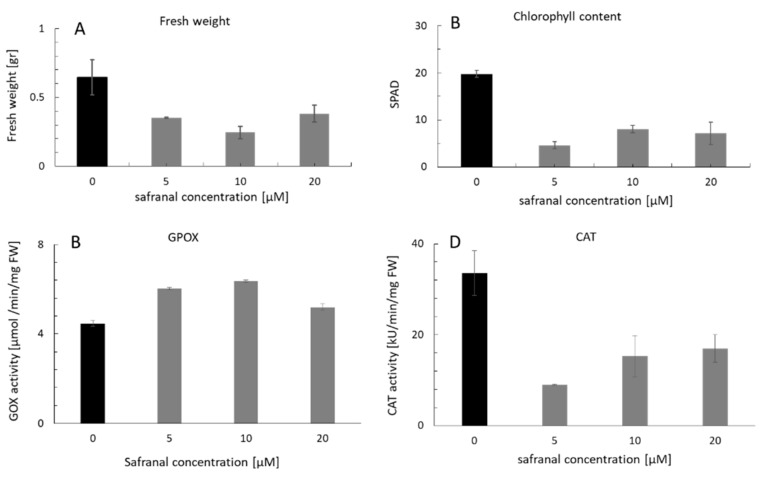
Physiological characteristics of *L. sativa* seedling after exposure to volatile safranal: (**A**) Fresh weight, (**B**) Chlorophyll content measured with a SPAD device, (**C**) (GPOX) activity and (**D**) CAT activity. Error bars indicate the mean values ± SD. Mean values are significantly different at *P* ≤ 0.05.

It is worth mentioning that chlorosis and necrosis similar to those caused by heat shock in plants were observed. In addition, at the maximum concentration of safranal, water burn shape spots were visible.

To assess the half maximal effective concentration (EC_50_) of safranal on CAT, the weight loss was considered as the final result of the continuous stress response of the test plant *L. sativa*. Therefore, the maximal weight decline caused by safranal was calculated by measuring its concentration in the headspace of the plant boxes using GC-MS. At the end of the experiment, the maximum CAT activity was observed in the control. However, it was followed by a decreasing pattern when safranal concentration increased. In addition, GC-MS analysis of the headspace gas in the designated plant boxes revealed that the EC_50_ on *L. sativa* seedling growth response was 6.12 µg/cm^3^ (Figure 3A,B). In Figure 3A,B, the application of 5 µM safranal aliquot of 6.12 µg/cm^3^ of safranal in its volatile form resulted in an approximately 50% decline in CAT activity and therefore a 50% weight loss. The result of this experiment indicates that the inhibition of oxidative enzymes such as CAT enzyme by safranal might be the main mechanism of action for its activity in the volatile form, indicating the uncompetitive inhibitory activity of safranal.

The current result was consistent with the previous report which indicated moderate inhibition of ROS scavenging enzymes by safranal on some mammalian cells [37]. It has also been reported that safranal inhibits the tubulin formation during cell division by binding to the oxygen molecule Gly 142 through a hydrogen bond [38], and therefore inhibits the cell division in cancer cells. However, its mechanism of action on the plants has not been well understood. Based on our observation, this result could be due to an increase in H_2_O_2_ and as a result of the uncompetitive nature of safranal on the CAT substrate active site. Comparing the structure of safranal to similar allelochemicals such as goniothalamin and trans-hexanal, an electrophilic α, the β-unsaturated carbonyl group is common as a putative bioactive moiety, indicating the involvement of this bioactive structure in the generation of oxidative stress [39] (Figure 4). Therefore, when compared to the results of this study, the formation of the hydrogen bond between the aldehyde group of safranal and the oxygen of the hem group of CAT enzyme can be the reason for the activity of safranal. Inhibition of the ROS scavenging enzymes by thousands of secondary metabolites and allelochemicals from plants that are produced and released into the natural ecosystem is known as their mechanism of action. Many of these compounds have an α,β-unsaturated carbonyl groups such as coumarin or juglone.

### 2.3. The Effect of Safranal on Growth and Physiological Characteristics of Weeds

In allelopathic studies, especially in agricultural science, the goal of finding a new allelochemical is often finding its potential in weed control purposes. Such studies also help to understand the effect of allelochemicals on other crops or to illustrate the allelochemical responsible for the particular phenomenon. Therefore, the effect of safranal on the growth and physiological characteristics of *L. sativa*, *T. pratense*, and *L. scoparius* (dicotyledonous), *D. Glomerata*, *P. pratense*, and *L. multiflorum* (monocotyledons) was studied to determine its possible herbicidal activity. It was observed that safranal significantly affected all the measured parameters. In the greenhouse experiment, leaves showed yellow coloration and necrosis at all surfaces, possibly due to the decline of chlorophyll content which could be considered as senescence caused by the allelopathic effect of safranal and ultimately causing plant death (Figure 5E).

The effect of safranal on physiological parameters of *L. sativa* and *L. scoparius* and *T. pratense* was significant (Figure 6). The chlorophyll content (SPAD) and fresh weight of *L. scoparius* and *T. pratense* decreased significantly as safranal application rate increased. Spraying 15 µM of safranal caused severe damage to the *T. pratense* leading to the death of the plants after 48 h. The results corroborate the findings for *L. sativa* seedlings seen in Figure 6. Although the *L. multiflorum* SPAD, leaf electrolytes leakage, and fresh weight were slightly affected by safranal, the effect was only significant on fresh weight when 15 µM safranal was used (Figure 6). In general, *L. multiflorum* showed resistance to sprayed safranal. A similar trend was observed for the monocotyledons species (*D. glomerata* and *P. pratense*). However, *P. pratense* showed significant tolerance against the cell membrane disruption activity of safranal. This observation is due to the structure and genetic makeup of monocots, a common phenomenon, which confers resistance to allelochemical and herbicides on them [40]. Similar to other allelopathic plants and allelochemicals in the inhibition of photosynthesis, an increase in ROS, chlorophyll decline, inhibition of enzymatic activity, and plant cell membrane disruption mechanisms have been widely reported [41,42]. Additionally, compared with other synthetic herbicides, safranal might be less harmful to humans given the fact that it has been identified in many fruits and plants and has been applied as an edible flavour and food-colouring agent over centuries [43,44]. In the present study, safranal showed stronger inhibitory activity against dicotyledonous in comparison to the monocotyledons. In addition, safranal initiated a significant weight and chlorophyll content reduction in all the tested plants.

## 3. Materials and Methods

In our previous studies, we reported a methodology that could be used to evaluate the growth inhibitory effects of volatile compounds on the early stage of seedling growth [14]. However, there is a dearth of information that the method could accurately measure, including the effective concentration of a volatile allelochemical on phenotypical or physiological changes of plants in later maturation stages. Developing a small-scale system that can estimate the concentration of volatile compounds and their associated response in plants is an essential need in the field of allelopathy. 

Therefore, a direct exposure technique for in vitro toxicity bioassay of volatile safranal was developed using cultured *L. sativa* seedlings grown on rockwool medium. The presented method not only escapes the current dreed in toxicity bioassay of VOCs but may also be a better simulation of exposure to volatile allelochemicals (e.g., safranal) in in vivo and on the filed condition.

As shown in Table 1, the organoleptic properties of safranal make it relatively easy to measure its physiological effect on test species in a static condition. Consequently, the required equipment needed for such test would be simple and easy. In this study, the method was initially developed and tested to measure the EC_50_ of safranal on *L. sativa* seedlings and then used for evaluation of the safranal effect on some common weed species.

### 3.1. The Allelopathic Activity of Safranal on Growth and ROS Scavenging Enzymes of L. sativa in Vitro Condition

#### Plant Material

*L. sativa* seeds (Great Lake 366, Takii Co., Kyoto, Japan) were selected as a test plant and grown in a 2 × 2 × 2 cm rockwool block in a growth chamber (16 h dark, 8 h light period). After 3 weeks of growth, single seedlings were transferred to a plant box container and cotton swabs containing 100 µL of different concentrations of safranal (0, 5, 10, and 20 µM (SAFC, ≥88%, Sigma-Aldrich, Bengaluru, India)) in Dimethyl sulfoxide, (WAKO, Osaka, Japan) were placed 1 cm next to them. Plant boxes were then air tightened and incubated for 48 h to allow safranal to evaporate and then the physiological response of the test plant including fresh weight, electrolyte leakage, catalase activity, and peroxidase activity were measured (Figure 7). The experiment was repeated four times with four replications. At the end of the incubation (48 h) and before using the seedlings for electrolyte leakage test, leaf chlorophyll content from the tired, true mature leaf of each three seedlings from each treatment was measured using a portable chlorophyll meter (SPAD-502Plus, Konica, Japan). The preliminary experiment indicated that the gas escape from the designated system was minor and did not affect the results.

### 3.2. Headspace Gas Chromatography-Mass Spectrometry (HS-GC-MS)

Headspace GC-MS was carried out to determine the EC_50_ of volatile safranal on the growth of *L. sativa* in the plant box. Quantitative analyses were established by a calibration curve with five different concentrations (1, 10, 50, 100, and 400 ppm diluted in hexane) of each constituent and the amount of constituent was calculated by the equation generated from the calibration curve. To assess the specific activity (EC_50_), the headspace of each treatment in the plant boxes was analyzed by HS-GC-MS. Into each plant box, 1000 μL of headspace gas was injected with a 5 mL SGE 5MDR-HSV syringe through a pre-designated septum installed on the plant box into the GC-MS QP5050 spectrophotometer equipped with Shimadzu GC 17A, EP5MS (5% phenyl methylsilane) capillary column (30 μm × 250 μm × 0.25 μm) and helium as a gas carrier. The operating conditions of GC-MS were as follows: The GC oven temperature was adjusted from 50–150 °C with an increase of 3 °C/min, held for 10 min, then raised to 200 °C with a rising temperature of 10 °C/min. The compounds were identified with mass spectra of by NIST spectra library. Mass spectra were recorded at 70 eV with a mass range of *m/z* 50 to 400, compared against an in-house mass spectral library (NIST and Wiley), and confirmed against the spectra of the authentic standard.

### 3.3. Quantification and Visualization of H_2_O_2_ Using DAB Staining Method

Visualization and quantification of H_2_O_2_ were performed according to previously described methods by Lee et al. [45] and Li et al [46], with some modifications. Week-old *L. sativa* seedlings were exposed to different concentrations of safranal (0, 5, 10, and 20 µM, SAFC, ≥88% Sigma-Aldrich) in a plant box container as described previously. After 48 h, the seedlings were cut from the collar and weighed. All samples were then placed in 10 mL polyethene falcon tubes containing 5 mL DAB solution (1 mg mL^−1^ DAB, Sigma catalogue number: D8001, pH 3.8 adjusted with HCl) with the caps removed in a vacuum desiccator and vacuum-infiltrated for 30 min at room temperature with DAB. Samples were then incubated in the dark overnight. The excess stain was washed with distilled water, and plant segments were placed on paper towels saturated with a fixative solution (ethanol: acetic acid, 3:1, *v/v*) in a petri dish and were kept in the dark for 24 h. All segments were washed off with distilled water to remove chlorophyll. Segments were transferred to paper towels saturated with water and left for 30 min to remove any fixative residues and then were photographed (Figure 2). Samples were kept in glycol a lactoglycerol solution (lactic acid: glycerol: H_2_O 1:1:1, *v/v/v*) for further examination. To evaluate the H_2_O_2_, *L. sativa* seedlings were vacuum-infiltrated with DAB (1 mg mL^−1^ DAB, Sigma pH 3.8 adjusted with HCl) for 10 min and then incubated in a shaker for another 1 h at 37 °C. The stain was pounded off, and chlorophyll was removed with ethanol (80%) for approximately 20 min. Plant segments were immediately homogenized in 1 mL of 0.2 M perchloric acid (HClO_4_) in a pre-chilled pestle and mortar and incubated in ice for 5 min. Samples were centrifuged (10,000× *g*, 10 min at 4 °C), and after collecting the supernatant, the absorbance was measured at 450 nm using spectrophotometry.

### 3.4. Catalase Activity

The measurement of catalase activity was based on the yellow-brown complex formation of undecomposed H_2_O_2_ with ammonium molybdate [47] with some modifications. One gram of plant material was grounded with 0.01% Polyvinylpolypyrrolidone (PVPP) and 1 mL of 0.1 M potassium phosphate buffer (pH 7.0) in a pre-chilled mortar and pestle. The mixture was then passed through a filter paper Number 1, centrifuged at 10,000 rpm at 4 °C and the supernatant collected as a protein extract. The reaction mixture was obtained by including 0.1 mL of supernatant in 0.4 mL of potassium phosphate buffer (60 mM, pH 7.0) containing 60 mM H_2_O_2_. The mixture was then incubated for 4 min at 34 °C, and the reaction was stopped by addition of 0.5 mL of 32.4 mM ammonium molybdate. Absorbance was recorded spectrophotometrically at 405 nm against a blank sample that contained all the reaction mixture.

Catalase activity (kU) was calculated using the following equation:
$$Catalase\;activity\left(kU\right)=\frac{A\left(\mathrm{Sample}\right)-A\left(\mathrm{blank}1\right)}{A\left(\mathrm{Sample}\right)-A\left(\mathrm{blank}3\right)\times 271}$$
Blank 1: 1.0 mL substrate, 1.0 mL molybdate and 0.2 mL sample;Blank 2: 1.0 mL substrate, 1.0 mL molybdate and 0.2 mL bufferBlank 3 contained:1.0 mL buffer, 1.0 mL molybdate and 0.2 mL buffer.

The results of this experiment were also compared with the methodology optimized by Hadwan et al. to increase the accuracy using the measured absorbance at 374 [48].

### 3.5. Visualisation of Superoxide (O-2) and Determination of Peroxidase Activity in L. sativa Seedlings

The Nitroblue tetrazolium (NBT) staining method was used to visualize the O^−^_2_ radicals in the plant leaves according to the method described by reference [49]. The seedlings were cut from the collar and then vacuum infiltrated with 0.5 mg mL^−1^ NBT for 2 min and incubated in the dark for 2 additional hours. Then, chlorophyll was removed by adding ethanol (96%): acetic acid (10:1) for 10 min. Photographs were taken using a stereomicroscope. The quantitative detection of ROS was performed as described by Jiang & Zhang [29]. Guaiacol peroxidase (GPOX; EC 1.11.1.7) activity was measured according to Zhou et.al [50] with minor modifications. The experiment was conducted based on the measuring of the GPOX scavenging activity by using guaiacol as a hydrogen donor. The reaction mixture contained 10 mM H_2_O_2_ in 50 mM phosphate buffer (pH 7), 9 mM guaiacol and the enzyme extract. GPOX activity was estimated by the increase in the absorbance of tetra-guaiacol at 470 nm and was expressed as micromole of guaiacol oxidised per min at 25 °C. The reaction mixture contained 2 mL of 100 mM sodium-acetic buffer (pH 5.4), 1 mL of 0.2 (*w*/*v*) guaiacol, 20 µl enzyme extract and 0.1 mL of 0.75% H_2_O_2_. GPOX activity was measured based on the increase in the absorbance at 470 nm as an indicator of polymerization of guaiacol.
The enzyme specific activity units g-1 FWt = [500/Δt] × [1/1000] × [TV/UV] × [1/FW]
where:Δt: Time changed in minTV: Total volumeUV: Volume usedFW: Fresh weight of leaf tissue

### 3.6. In Situ Effect of Safranal on Growth and Physiological Characteristics of L. sativa and Some Common Weeds

#### 3.6.1. Plant Material

Based on our primary observation and to understand some aspects of the mechanism of action of safranal, its effect on growth and physiological characteristics of *L. sativa* and some common weeds (three-week-old seedlings) was investigated. The physiological parameters and antioxidant enzyme activities of *L. sativa* exposed to the allelochemical safranal were assayed in a separate experiment. Seeds of weeds, *Lotus scoparius, Trifolium pratense*, *Dactylis glomerata*, *Phleum pratense*, and *Lolium multiflorum* were purchased from Benidai Company, Japan, while *L. sativa* (Great Lake 366) seeds were purchased from Takii Co., Japan. Seeds were sown and grown in 2 × 2 × 2 cm rock wool block medium in an incubator (16 h dark, 8 h light period). Following a period of 3 weeks growth, seedlings were transplanted to 10 × 10 plastic pots and placed in a greenhouse. After 1 week of acclimation, plants were sprayed with different concentrations of safranal (0, 6, and 15 µM) suspended in distilled water using 1% laboratory detergent. The control was only sprayed with 1% laboratory detergent solution. The fresh weight, electrolyte leakage, and chlorophyll content were recorded immediately after plantlets were removed from the pots (after 48 h).

#### 3.6.2. Measurements of Fresh Weight, Chlorophyll Content, and Electrolyte Leakage

The plant leaves electrolyte leakage were measured as described previously by Zhu et al. and Pilloff et al. [51,52]. The weight of each seedling was measured, and immediately leaf discs were cut with a custom disc cutter (0.5 cm in diameter). Each disc was put into a beaker containing 20 mL of deionized water and washed slowly using a rotary shaker (100 rpm) to remove any solutes or contamination of damaged leaf. Three leaf discs from each replication were suspended in 5 mL of distilled water. Suspended leaf discs were then shaken at room temperature for 10 min, and the 2 mL liquid sample was immediately transferred to a compact conductivity meter (B-771, LAQUAtwin, HOBIRA), and Electrolyte Conductivity (EC) was recorded. Before the electrolyte leakage measurements, the leaves chlorophyll content from the same leaf of three selected seedlings was measured using a portable chlorophyll meter (SPAD-502Plus, Japan).

### 3.7. Data Analysis

Enzyme inhibitory assays were conceded based on three separate times. Measurements were taken in triplicates. Data shown in this research are expressed as mean ± standard deviation. Statistical analysis was performed using ANOVA to compare the means of different elements. Mean values are significantly different at *P* ≤ 0.05.

## 4. Conclusions

This study was carried out to determine the effect of safranal, the main allelochemical of Saffron volatiles, on the growth of *L. sativa* and fluctuations in enzymatic activities concerning its dose-response. Safranal affected *L. sativa* seedling growth parameters and inhibited the CAT activity at concentrations of 6.12 µg/cm^3^ (EC_50_) when it was applied in its volatile form. Safranal significantly inhibited the CAT activity but had a stimulatory activity on GPOX, the two major known ROS scavenging enzymes. Moreover, the designed methodology in this study showed a promising result for screening purposes of volatile allelochemicals on the growth and survival of other plant species.

A significant decline in chlorophyll content and fresh weight of *L. sativa* seedlings were observed. The morphologic response of the test weed species showed significant sensitivity to safranal toxicity. The results of evaluating the toxicity of safranal on the growth parameters of common weeds are comparable with those of *L. sativa* CAT activity and its dose-response. Therefore, incompetitive enzyme inhibitory activity of safranal on CAT and severe oxidative damage could be the main mechanism of action of safranal in plants. Observation of the growth characteristics of plants might be appropriate for evaluating the toxicity of safranal on other crops or weeds. However, supporting experiments such as transcriptomics and next-generation sequencing may be needed to further accentuate the results of this work. The relatively static generation of test atmospheres of VOSs (i.e. safranal) requires relatively simple equipment and procedures, making this method ideal for screening purposes. Thus, the developed method in this experiment could be applied for the evaluation of allelopathic activity of volatile allelochemicals.

## Figures and Tables

**Figure 1 molecules-24-01788-f001:**
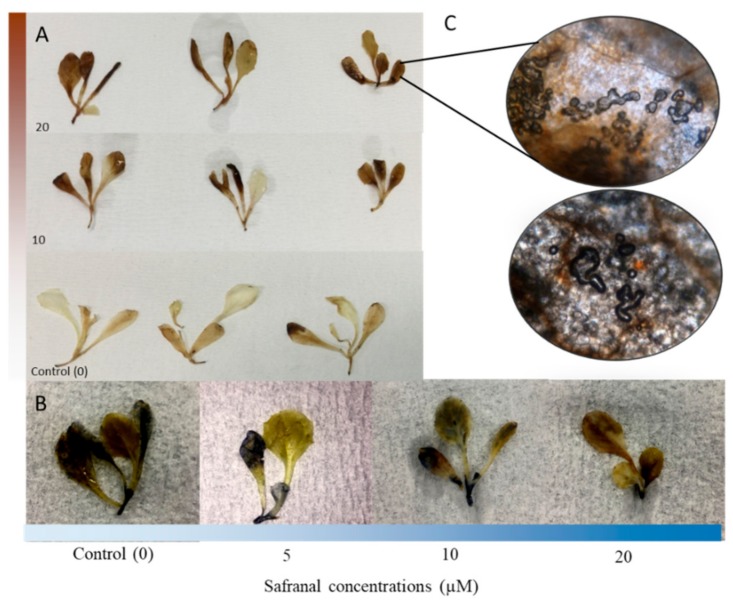
Visualization of superoxide and hydrogen peroxidase accumulation and cell/tissue disruption in *L. sativa* plants due to volatile safranal: Visualization of superoxide (**A**), hydrogen peroxidase accumulation (**B**) and cell/tissue disruption (**C**).

**Figure 3 molecules-24-01788-f003:**
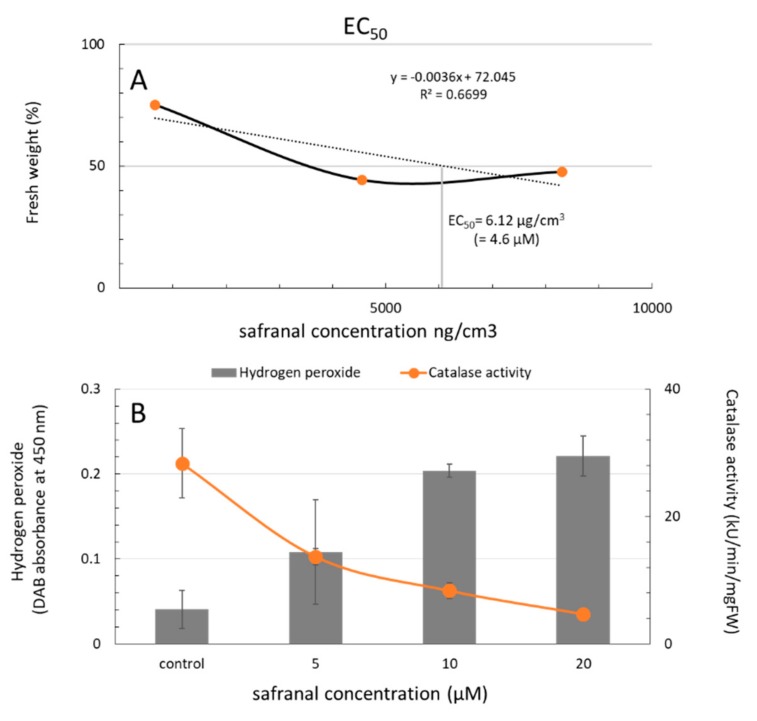
The effective concentration EC_50_ of volatile safranal (equivalent to ng/cm^3^) on the weight of *L. sativa* seedlings in plant box and (**B**) The effect of safranal on the activity of CAT enzyme and generation of reactive oxygen species (ROS): (**A**). Error bars indicate the mean values ± SD. Mean values are significantly different at *P* ≤ 0.05.

**Figure 4 molecules-24-01788-f004:**
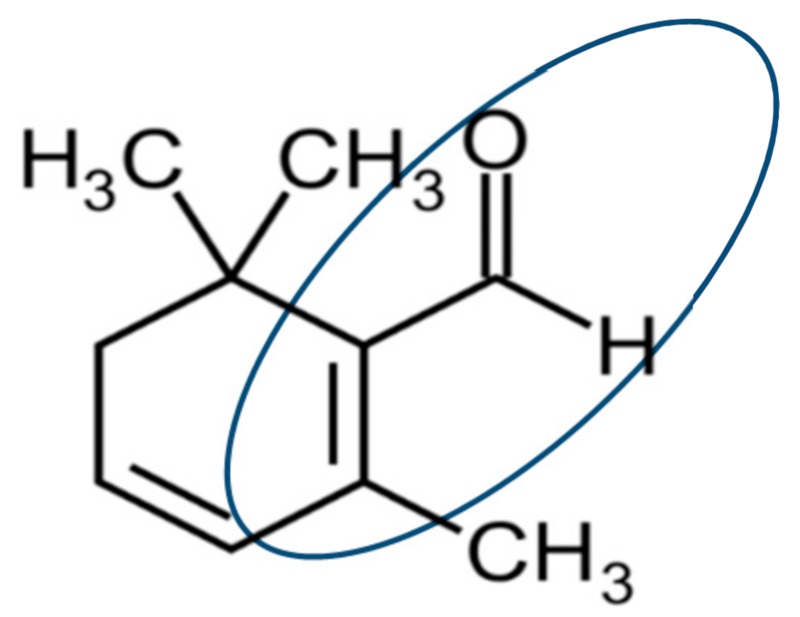
Structure of safranal and its active site.

**Figure 5 molecules-24-01788-f005:**
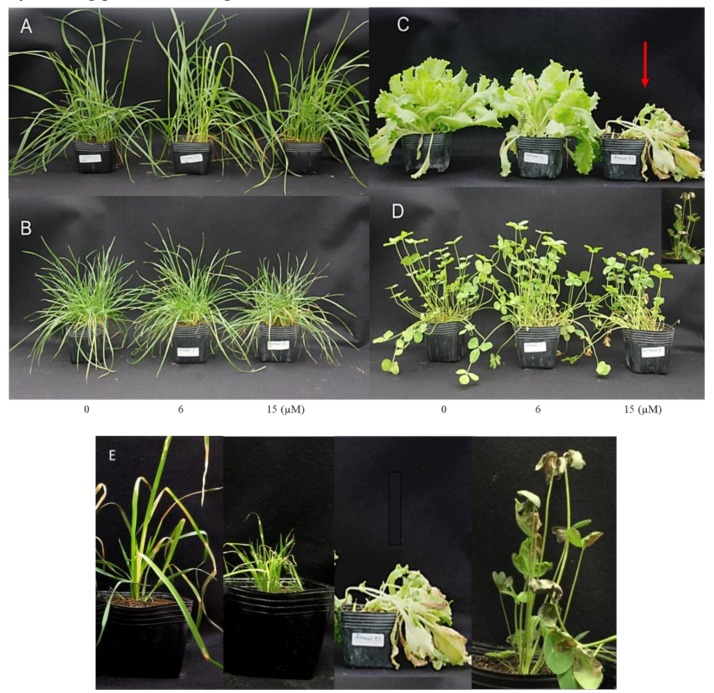
Yellow coloring and necrosis at all leaves surfaces and moving downwards (senescence indicted by arrow) in *D. glomerata* (**A**), *L. perenne* (**B**) *L. sativa* (**C**), and *T. repens* (**D**) due to a decline in chlorophyll content caused by safranal. Arrow indicates the severe damage of safranal on *L. sativa* leaves. (**E**) Yellow coloration and necrosis on tested species caused by safranal.

**Figure 6 molecules-24-01788-f006:**
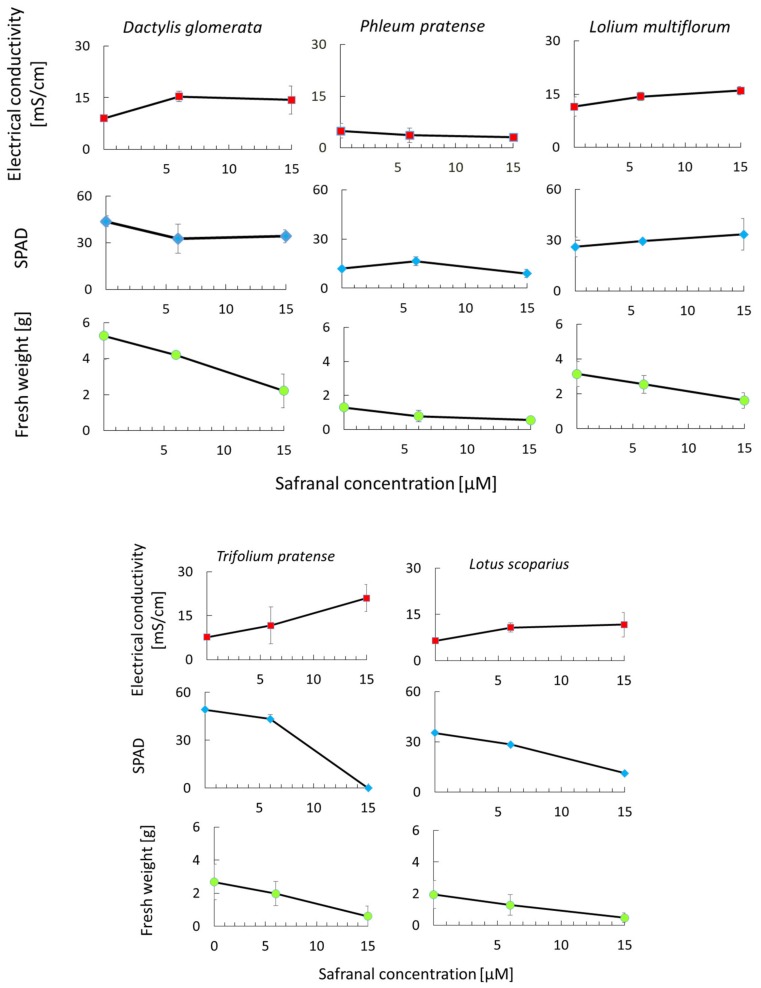
The effect of safranal on the growth and physiological parameters of monocots (*D. glomerata, P. pratense, L. multiflorum*) and dicots (*T. pratens, L. scoparius* ) weeds. *P* ≤ 0.05.

**Figure 7 molecules-24-01788-f007:**
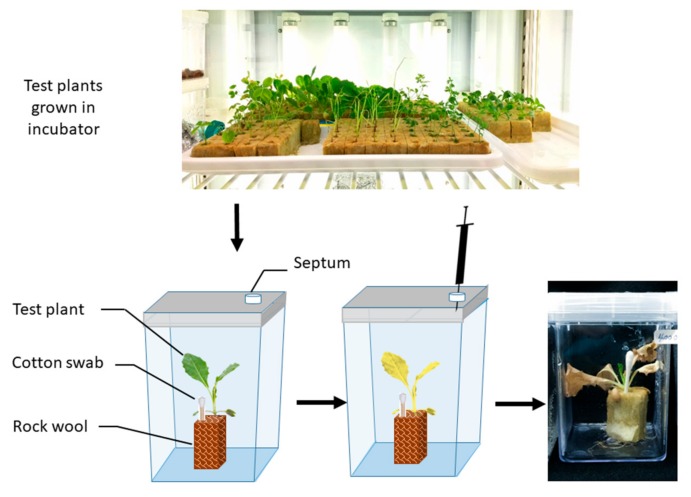
Schematic of the gas chamber growth for estimation of volatile compound EC_50_.

**Table 1 molecules-24-01788-t001:** Organoleptic properties of safranal.

Organoleptic Properties	Safranal (CAS No 116-26-7)
Appearance	Pale yellow to Yellow liquid
Molecular Formula	C_10_H_14_O
Molecular Weight	150.22
Specific Gravity	0.950–0.970 at 25 °C
Vapor Pressure	0.134 mm/Hg at 25.0 °C. (est)
Water solubility	Insoluble

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
