# Peer review of "Evaluation of Biological Response of Lettuce (Lactuca sativa L.) and Weeds to Safranal Allelochemical of Saffron (Crocus sativus) by Using Static Exposure Method"

_molecules, 2019, doi:10.3390/molecules24091788_

Round 1
Reviewer 1 Report
This paper is well written and data will be useful for organic weed control.

Author Response
Dear reviewer
We would like to thank the reviewer for careful and thorough reading of this manuscript and for
the thoughtful comments and constructive suggestions. your suggestions help to improve the quality of this manuscript. we have responded the suggestions and comments in the attached files. changes have been emphasized in the body of the manuscript in red color (the reviewer’s comments are in italics).
Best regards

Reviewer 2 Report
In the MS "Evaluation of biological response of Lettuce (Lactuca sativa L.) and weeds to safranal allelochemical of Saffron (Crocus sativus) by using static exposure method, authors investigated allelopathic potential of Saffron and identified safranal as inhibitory agent. They also investigated mode of action of safranal in order to its potential use as herbicide. Through biochemical assays and bioassays in greenhouse, they effectively proved that safranal can be used herbicide, however only drawback in this study that they did not include any field data. Considering novelty of this work, I would recommend this article for publication.
Author Response
We would like to thank the reviewer for careful and thorough reading of this manuscript and for the thoughtful comments and constructive suggestions, which help to improve the quality of this manuscript. Our response follows:
In this study, we focussed on the phytotoxicity of safranal, the main volatile allelochemical Saffron (Crocus sativus) against weed species and test plant lactuca sativa in the laboratory and greenhouse condition. Safranal showed the potential to inhibit the growth of the weed species in both in vitro and greenhouse experiments. Moreover, safranal was found to significantly suppress the activity of catalase, an important reactive oxygen species scavenging enzyme for the first time. Since there is the urgent need to innovate or improve the existing weed control strategies, we believe the information provided on safranal and its allelopathic effect will be valuable for further research. We believe the information provided by our experiments is need to be published as soon as possible so that the field scientist and practitioners could benefit the results of this work. Although we also believe that the field experiment is necessary for safranal, field experiment of the large scale usage of safranal is now undergoing in our group.

Reviewer 3 Report
This work begins the study of the possible mechanism of action of safranal, the main volatile chemical substance of saffron, to estimate its possible allelopathic effect. Just a few comments:
Line 45 and 46: some punctuation errors.
Some errors in the order of the figures: line 88 names figure 2, before figure 1 that is named in line 109. The same happens with figure 4 (line 157) and figure 3 (line 172).
Line 244: EC50 instead of EC50
Author Response
Dear reviewer
We would like to thank the reviewer for careful and thorough reading of this manuscript and for the thoughtful comments and constructive suggestions, which help to improve the quality of this manuscript. We have responded to the comments and suggestions in the attached file (the reviewer’s comments are in italics). Moreover, we have indicated the changes in the body of the manuscript using red color.
Best regards
